# High-throughput screening unveils nitazoxanide as a potent PRRSV inhibitor by targeting NMRAL1

Zhanding Cui [1], Jinlong Liu[1], Chong Xie[2], Tao Wang[1], Pu Sun[1], Jinlong Wang[1,3], Jiaoyang Li[1], Guoxiu Li[1], Jicheng Qiu[4], Ying Zhang[5], Dengliang Li[6], Ying Sun[1,7], Juanbin Yin[1], Kun Li[1], Zhixun Zhao[1], Hong Yuan[1], Xingwen Bai[1], Xueqing Ma[1], Pinghua Li[1], Yuanfang Fu[1], Huifang Bao[1], Dong Li[1], Qiang Zhang[1], Zaixin Liu[1], Yimei Cao[1,8] ✉, Jing Zhang [1,8] ✉ & Zengjun Lu [1,8] ✉

Porcine Reproductive and Respiratory Syndrome Virus (PRRSV) poses a major threat to the global swine industry, yet effective prevention and control measures remain elusive. This study unveils Nitazoxanide (NTZ) as a potent inhibitor of PRRSV both in vitro and in vivo. Through High-Throughput Screening techniques, 16 potential anti-PRRSV compounds are identified from a library comprising FDA-approved and pharmacopeial drugs. We show that NTZ displays strong efficacy in reducing PRRSV proliferation and transmission in a swine model, alleviating viremia and lung damage. Additionally, Tizoxanide (TIZ), the primary metabolite of NTZ, has been identified as a facilitator of NMRAL1 dimerization. This finding potentially sheds light on the underlying mechanism contributing to TIZ's role in augmenting the sensitivity of the IFN-β pathway. These results indicate the promising potential of NTZ as a repurposed therapeutic agent for Porcine Reproductive and Respiratory Syndrome (PRRS). Additionally, they provide valuable insights into the antiviral mechanisms underlying NTZ's effectiveness.

Porcine Reproductive and Respiratory Syndrome (PRRS), initially identified in the United States in the 1980s, rapidly proliferated throughout Europe in the subsequent years[1–3]. By approximately 2006, a major outbreak of PRRS had severely impacted China's swine industry, leading to substantial economic repercussions[4]. A previous study quantifies the significant financial burden of PRRS, estimating the cost at around 1424.37 yuan per sow in China, 126 euros in Europe, and 121 dollars in the United States[5]. The causative agent,

Porcine Reproductive and Respiratory Syndrome Virus (PRRSV), presents a formidable challenge to the global swine industry, predominantly inducing reproductive disorders in sows and acute respiratory diseases[6]. The mortality rate in cases involving highly virulent strains of PRRSV can escalate to 100%[7]. Investigations into PRRSV have revealed intricate interactions with the host immune system[6]. As a member of the *Arteriviridae* family within the *Nidovirales* order, PRRSV is noted for its highly mutable genome, which

[1]State Key Laboratory for Animal Disease Control and Prevention, College of Veterinary Medicine, Lanzhou University, Lanzhou Veterinary Research Institute, Chinese Academy of Agricultural Sciences, Lanzhou 730000, China. [2]Institute of Pathology, University Medical Center Göttingen, Göttingen, Germany. [3]Institute of Traditional Chinese Veterinary Medicine, College of Veterinary Medicine, Gansu Agricultural University, Lanzhou, China. [4]National Center for Veterinary Drug Safety Evaluation, College of Veterinary Medicine, China Agricultural University, Beijing 100193, China. [5]The Affiliated Animal Hospital of Jinzhou Medical University, Jinzhou, China. [6]Veterinary Immunology Laboratory, College of Veterinary Medicine, Northwest Agriculture and Forestry University, Yangling, Shaanxi Province 712100, China. [7]College of Veterinary Medicine, South China Agricultural University, No483 Wushan Road, TianheDistrict, Guangzhou 510642, China. [8]Gansu Province Research Center for Basic Disciplines of Pathogen Biology, Lanzhou 730046, China. ✉e-mail: caoyimei@caas.cn; zhangjing@caas.cn; luzengjun@caas.cn

facilitates frequent recombination events and results in a diverse spectrum of genotypes[8].

The PRRSV has precipitated a global pandemic, impacting regions in the Americas, Europe, and Asia. Present control strategies primarily utilize inactivated and attenuated vaccines[9]. However, despite intensive global efforts towards vaccine development, inactivated vaccines have not been able to induce sufficiently potent neutralizing antibodies, and attenuated vaccines have raised safety concerns[10]. Recent research underscores the limitations of current commercial PRRSV vaccines, as exemplified by an outbreak in China involving the NADC-30-like strain in vaccinated swine[11,12]. In response to these challenges, Holtkamp, in 2011, proposed a classification system for swine herds based on their viral status, which was later revised in 2021[13,14]. According to Holtkamp's model, ~30% of swine populations are classified in a Positive Unstable (I) state[15]. The complexity of the existing control measures, which rely predominantly on vaccines, public health interventions, or gene-edited swine, suggests that national eradication of PRRSV remains a formidable challenge[16].

Antiviral medications are pivotal in controlling viral infections and are integral to pandemic management strategies. However, the development process for traditional antiviral drugs is characteristically protracted[17]. In this context, employing High-Throughput Screening (HTS) for the reevaluation of existing drugs and the swift identification of new candidates has emerged as a highly efficient method in the treatment of viral diseases[18]. HTS primarily encompasses two approaches: phenotypic-based and target-based screening[19]. The complexity of target-based screening lies in unraveling the intricacies of viral life cycles, pinpointing precise potential targets, and determining accurate protein structures—tasks that are particularly challenging in veterinary medicine due to the wide array of pathogens involved. Conversely, phenotypic screening presents its own set of challenges, notably in the development of suitable viral reporting systems, often necessitating support from reverse genetics techniques. Once promising lead compounds are identified, the subsequent phase involves determining their specific sites of action to unravel the mechanisms underlying their antiviral effects[19].

In this research, we initially developed a Green Fluorescent Protein (GFP)-based cell line to report PRRSV infection and conducted a screening of 16 potential anti-PRRSV compounds from an extensive FDA Approved & Pharmacopeial Drug Library, which contains 3274 compounds. Nitazoxanide (NTZ) emerged as a significant inhibitor of PRRSV proliferation, both in vitro and in vivo. Our findings reveal that NTZ administration, whether applied prophylactically or therapeutically and at a specific dosage, effectively reduces viremia and lung pathology in infected swine, thereby diminishing viral shedding and curtailing the transmission of PRRSV among the animal population. Furthermore, NTZ was found to interact with multiple target proteins. Notably, its primary metabolite, Tizoxanide (TIZ), has been identified as inducing the dimerization of NMRAL1, thereby enhancing the response of Interferon Beta (IFN-β) in the post-infection stage of PRRSV.

## Results

### High-throughput screening targeting PRRSV
In our study, we opted for a phenotypic-based approach to design a HTS model targeting PRRSV. Given the compact nature of the PRRSV genome, inserting GFP or Luc genes could affect genome stability[20]. Therefore, we established a cell line capable of reporting PRRSV (the GSWW-18 strain) infection (Fig. 1a). Marc-145-GFP cells expressed GFP protein post-PRRSV infection (Fig. 1b) and exhibited specificity (Fig. 1c). Upon infection of Marc-145-GFP cells with either the GSWW-15 or VR-2332 strains, green fluorescence emerged at the cytopathic effect sites (Supplementary Fig. 1a). Specifically, we co-introduced virus (Multiplicity of Infection, MOI = 1) and compounds into a 96-well plate, inoculating 1000 cells per well (Fig. 1d). We assessed the antiviral

potential of 3274 compounds from the FDA approved and pharmacopeial drug library in Marc-145-GFP reporter cells against PRRSV. Our HTS model achieved a Z′ factor of 0.724 (Fig. 1e, Supplementary Fig. 1b) and an average signal-to-noise (S/N) ratio of 381 (Supplementary Fig. 1c), indicating this method's efficacy in identifying potential antiviral compounds. Next, we selected the highest percentage of fluorescent cells in NC (Negative Control) and RBV (Ribavirin positive control) groups as the upper limit for antiviral activity (Supplementary Fig. 1d) and the average cell count in the RBV group as the lower limit for drug toxicity (Supplementary Fig. 1e). In the first round of screening, we identified 141 potential antiviral compounds (Fig. 1e). Gene Set Enrichment Analysis (GSEA) revealed significant enrichment of these compounds' targets in 5-HT, Dopamine Receptor, Serotonin Transporter, Toll-like Receptor, etc. (Fig. 1f). Gene Ontology analysis also linked these targets with multiple signaling pathways, including Neuroactive ligand-receptor interaction, Mitogen-activated Protein Kinase (MAPK) signaling pathway, and FoxO signaling pathway, etc (Supplementary Fig. 1f). These data suggest a possible association between PRRSV proliferation and these receptors or pathways.

### Evaluation of selected compounds for antiviral activity
Subsequently, a second-round screening of the selected 141 compounds was conducted on Marc-145 cells using immunofluorescence assay (IFA) at four different dosage levels (Fig. 2a; Supplementary Fig. 2). Seventeen of these compounds demonstrated significant antiviral efficacy at 5 μM (Supplementary Fig. 2). Cluster analysis of antiviral effects led to the identification of the top 17 compounds that maintained substantial antiviral activity at 5 μM (Fig. 2b; Supplementary Fig. 2), without exhibiting common drug targets (Fig. 2b). In the third round of screening, after excluding Deslanoside due to its high cytotoxicity, the $EC_{50}$ and cytotoxicity of the remaining 16 compounds were assessed (Fig. 2c). Chloroquine, Acetophenazine, Pizotifen, Terbinafine, Remdesivir, Benzydamine, Hydroxychloroquine, and NTZ showed low cytotoxicity. With its affordable cost and an $EC_{50}$ in the sub-micromolar range, NTZ's efficacy in inhibiting PRRSV infection has attracted considerable attention. This interest is further amplified by our previous studies, which demonstrated NTZ's ability to inhibit *Feline Calicivirus* (FCV) infection[21]. The cytotoxic concentrations ($CC_{50}$) of NTZ was around 44 μM in both Porcine alveolar macrophages (PAM) and MARC-145 cells after 48 h of exposure (Supplementary Fig. 3a). The $EC_{50}$ of VR2332 and GSWW-15, strains preserved in our lab, was ~0.67 μM and 1.17 μM (Supplementary Fig. 3b). There was a negative correlation between NTZ concentration and viral titers of all three strains (Supplementary Fig. 3c), and IFA on PAM cells also indicated a dose-dependent inhibitory effect of NTZ on PRRSV (Supplementary Fig. 3d).

### Efficacy of NTZ against PRRSV in swine
The antiviral activity of NTZ in swine infected with the GSWW-18 strain was further assessed as described in Method In Vivo Antiviral Effect of NTZ. Animals were assigned to pre-infection treatment at −1 days post infection (dpi) to evaluate the prophylactic efficacy of NTZ, and post-infection treatment at 3 dpi to assess therapeutic effects (Fig. 3a). Clinical scores were documented on day 7, and one pig from each group was euthanized on day 14 for organ harvest and subsequent analyses (Fig. 3a). Post-PRRSV infection, severe interstitial pneumonia with lobar consolidation was noted (Fig. 3b), which was significantly mitigated by a specific dose of NTZ (Fig. 3b). Immunohistochemistry (IHC) revealed that oral administration of 25 mg/kg NTZ in both prophylactic and therapeutic groups resulted in fewer lung PRRSV-positive cells compared to MOCK-treated controls (Fig. 3c). Starting from day 0 of infection, daily monitoring included body temperature, weight, collection of oronasal and anal swabs, and blood sampling (Supplementary Fig. 4a). PRRSV was predominantly localized in the lungs, mesenteric lymph nodes (MesLNs), mandibular lymph nodes

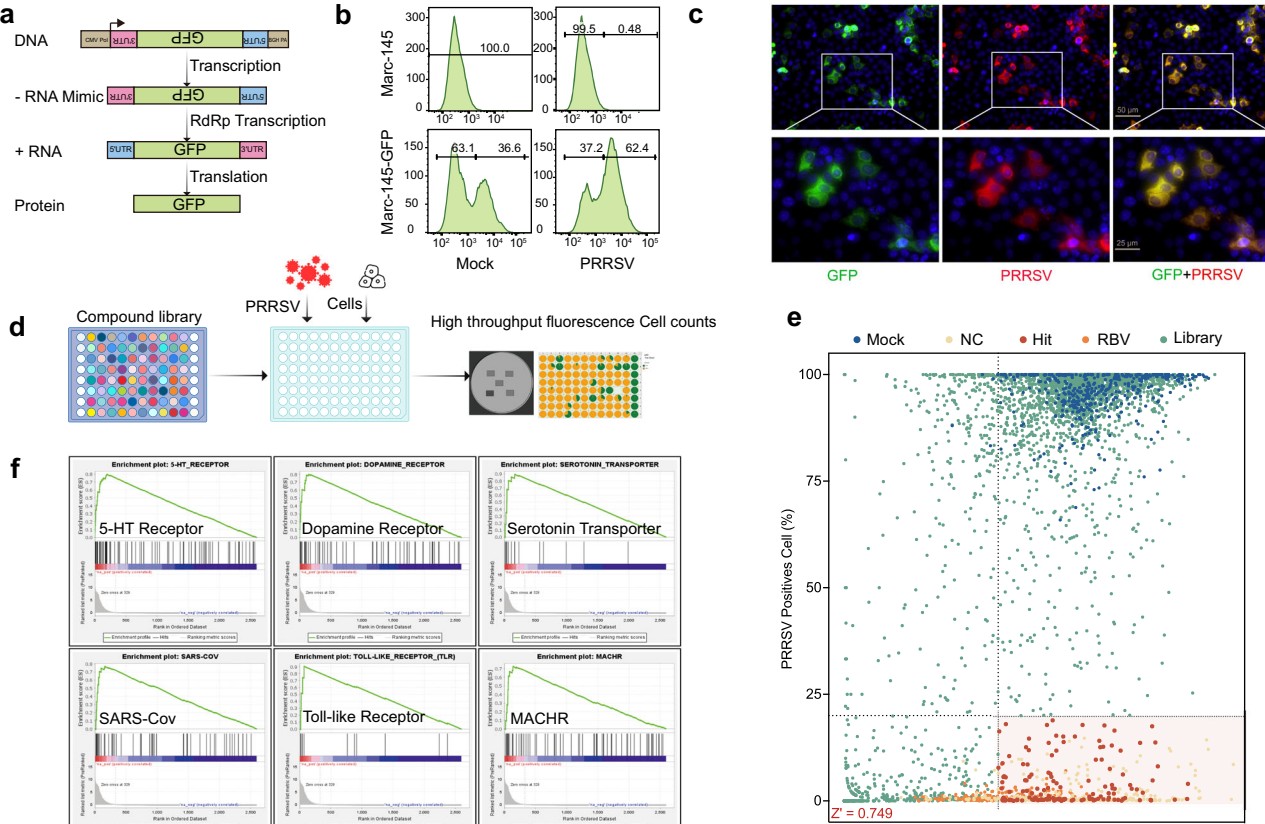

**Fig. 1 | High-throughput screening for antiviral compounds against PRRSV.**
**a** Strategy diagram for GFP expression post-PRRSV infection in cells. Following PRRSV infection, the virus encounters cellular negative-strand PRRSV genome analogs, converting them into positive-strand RNA, subsequently expressing GFP. **b** Flow cytometry diagram of GFP expression induced in Marc-145-GFP cell line post-PRRSV infection. GFP-expressing cells increased by 25.8% in Marc-145-GFP cells post-PRRSV infection. **c** Indirect immunofluorescence image of Marc-145-GFP cells post-PRRSV infection. Cells infected with PRRSV expressed GFP in conjunction with PRRSV-N protein within the same cells. **d** Strategy diagram for implementing HTS. Compounds, GSWW-18 (Multiplicity of Infection, MOI = 1), and Marc-145-GFP cells were co-introduced into 96-well plates. The first column of the plate was for Mock treatment, with the addition of equal concentration Dimethyl sulfoxide (DMSO) and virus, and the last column contained 60 μM Ribavirin. **e** Raw data from all screening plates. DMSO treatment group (Mock, blue), negative control group (NC, yellow), potential hits (Hit, red), 60 μM Ribavirin positive control group (RBV, orange), compound library (Library, green). Data represent the average of three to five images per well. **f** Identification of enriched targets for potential antiviral compounds through GSEA. Compounds were ranked according to antiviral activity, and enrichment scores were then ranked, with the top 6 targets shown in the diagram ($P < 0.05$, false discovery rate (FDR) $q < 0.5$). $P$-values were generated using a one-sided hypergeometric test. Each black line in the diagram represents a compound.

(ManLNs), inguinal lymph nodes (ILNs), and tonsils, with viral presence also detected in other organs (Supplementary Fig. 4b, c). Oral administration of 25 mg/kg NTZ alleviated the rise in body temperature and weight loss in infected animals (Supplementary Fig. 5). NTZ significantly reduced PRRSV replication in ManLNs, lungs, and tonsils in both prophylactic and treatment groups ($P < 0.0332$, Fig. 3d or f). Notably, only the 25 mg/kg dose significantly suppressed the virus in ILNs in the treatment group ($P < 0.0001$). In both the treatment and prophylactic groups, the 25 mg/kg group exhibited no apparent clinical symptoms by day 7 and no fatalities occurred within 14 days (Fig. 3e). In the bronchoalveolar lavage fluid (BALF), significant effects were observed in the prophylactic group at both the 10 mg/kg and 25 mg/kg dosages ($P < 0.0001$, Fig. 3g). However, in the treatment group, only the 25 mg/kg dosage achieved statistical significance ($P < 0.0001$, Fig. 3g). Ribavirin (RBV) appeared to lack efficacy against PRRSV in swine. No adverse reactions to NTZ were observed in pigs in terms of hematology, biochemical indices, and clinical activity following three consecutive days of high-dose oral administration over an 8-day period (Supplementary Fig. 6).

**NTZ inhibits PRRSV transmission**
Analysis of swabs and blood samples showed peak PRRSV viral copy numbers in blood and oronasal swabs occurring between days 2 and 4,

and in anal swabs between days 2 and 5 (Supplementary Fig. 7). In the prophylactic group, 5 mg/kg NTZ significantly reduced PRRSV viral load in blood ($P = 0.0033$), and a 10 mg/kg dose significantly lowered oronasal shedding ($P = 0.0012$; Fig. 4a). However, even at 25 mg/kg, NTZ did not reduce anal shedding. In the treatment group, 10 mg/kg NTZ significantly decreased both blood viral load and oronasal shedding ($P < 0.0001$), but did not affect anal and oronasal shedding (Fig. 4a). RBV was effective only in the prophylactic setting (Fig. 4a). Although the replication mechanism of PRRSV in blood is not fully understood, viremia is an undoubted marker of disease severity[22,23]. It was then investigated whether NTZ directly inhibits PRRSV proliferation in blood. On the second day post-infection, we administered NTZ at dosages of 10 mg/kg or 5 mg/kg and monitored the dynamics of PRRSV in the blood within 24 h (Fig. 4b). We observed that NTZ acted swiftly. With the 5 mg/kg dosage, viral loads decreased from 2 h to 8 h post administration, rebounded at 12 h before the second dose, and were suppressed again after re-administration; the 10 mg/kg dosage showed more pronounced inhibition (Fig. 4b), indicating direct suppression of PRRSV proliferation in blood by NTZ. Further experiments explored NTZ's role in viral transmission (Fig. 4c). Except for two uninfected co-house animals in the 25 mg/kg dose group, all in the 10 mg/kg group contracted PRRSV (Fig. 4d, e), suggesting that 25 mg/kg NTZ can help limit PRRSV transmission. Pharmacokinetics of

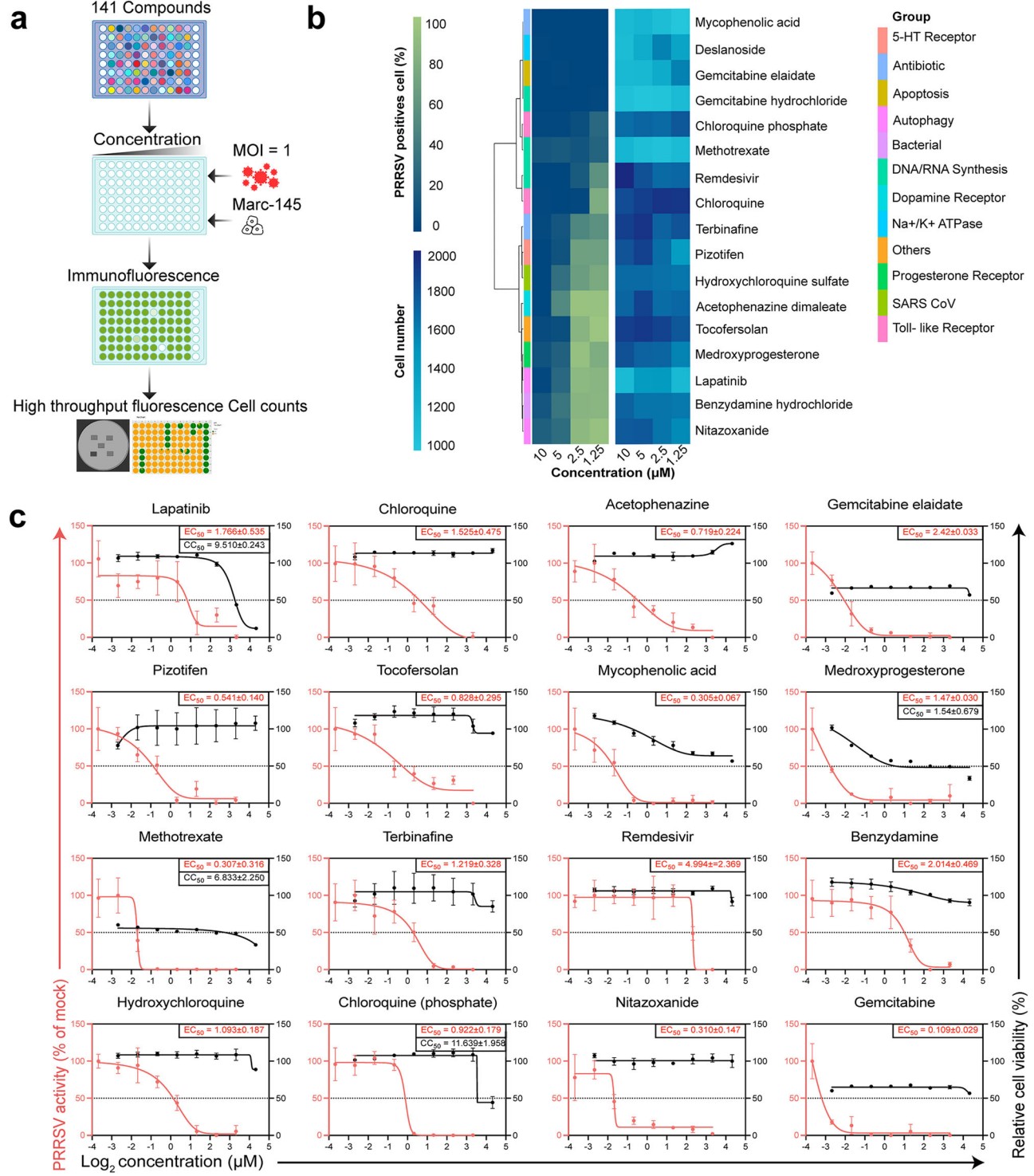

**Fig. 2 | The screening and dose-response relationship of hit compounds.**
**a** Scheme of screening compounds. A concentration-gradient screening strategy was employed for 141 compounds using indirect immunofluorescence assay (IFA). Compounds were first diluted in cell culture medium to 20 µM and subsequently subjected to twofold serial dilutions, before co-incubation with Marc-145 cells and GSWW-18 strain in a 96-well plate. **b** The top 17 compounds from the cluster analysis of 141 compounds are presented. Compounds were clustered based on their targets using the complete linkage method and Euclidean distance. **c** Dose−response and cytotoxicity of the 16 compounds. Half-maximal effective concentration ($EC_{50}$) curves (in red, $n = 2$ biologically independent samples) and half-maximal cytotoxic concentrations ($CC_{50}$) curves (in black, $n = 3$ biologically independent samples) for the different compounds were determined in PAM cells. Data are mean ± SEM.

Tizoxanide (TIZ; NTZ's main metabolite) in swine revealed short half-life and peak drug times for NTZ (Fig. 4f; Supplementary Fig. 8). TIZ was primarily distributed in small and large intestines, MesLNs, ILNs, but not in high concentrations in lungs (Fig. 4g).

## Identifying targets for NTZ action

The pronounced antiviral phenotype led us to explore the mechanism behind NTZ's inhibition of PRRSV proliferation. We adopted a classic approach of administering the drug at various time points to study

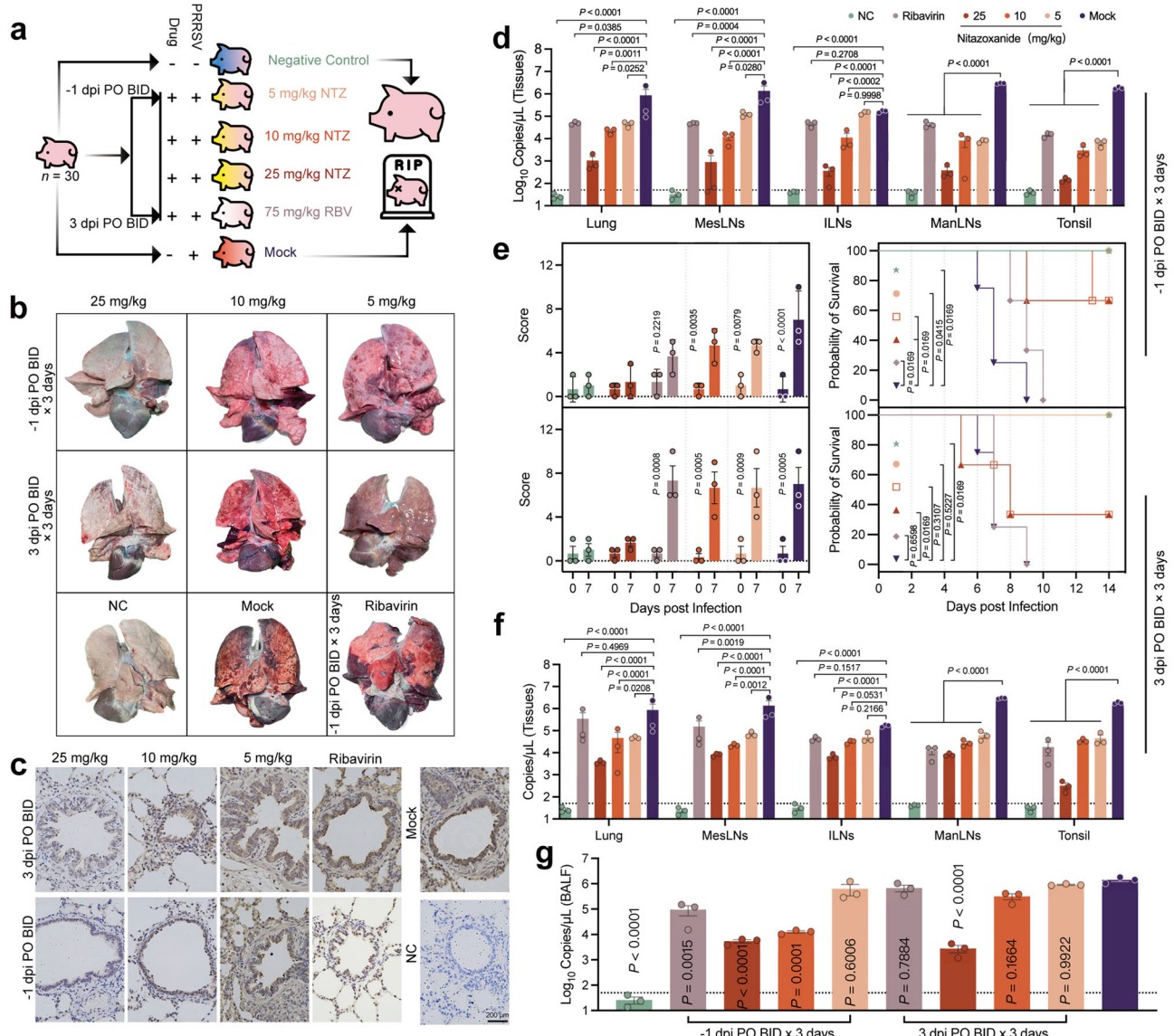

**Fig. 3 | In vivo antiviral efficacy of NTZ. a** Animal grouping schematic. Pigs were intranasally infected with $1 \times 10^{5.5}$ TCID$_{50}$ of GSWW-18 strain. NTZ was weighed according to animal body weight, suspended in PBS, and orally administered using a gavage. Mock treatment (Mock) groups received an equivalent volume of PBS orally. Prophylactic group received NTZ orally one day prior to GSWW-18 infection, and treatment group received NTZ on the third day post-infection. NTZ was administered orally (PO) twice a day (BID) for three consecutive days. Animals ($n = 3$) in each group were housed separately to prevent cross-infection, with caretakers changing protective clothing and shoe covers when cleaning enclosures and feeding. **b** Representative lung images. Mock group samples were collected at the time of the last pig's death. **c** Representative IHC images of lung tissue. Brown granules indicate PRRSV positive (primary antibody SR30). The images are

representative of three independent biological replicates. **d** Viral load in representative tissues from the prophylactic group. **e** Clinical scores (left) and survival curves (right) for both prophylactic and treatment groups. Clinical scores for all animals were compiled on days 0 and 7. Colors correspond to groups in Fig 3d. **f** Viral load in representative tissues from the treatment group. **g** Viral copy numbers in BALF. Dashed line indicates the limit of detection (**d, f, g**). Symbols represent three independent biological replicates (**d–g**). Bar graphs (**d, e, f, g**) show mean ± SEM values of samples, with *P*-values as indicated, calculated via two-way ANOVA with Dunnett's multiple-comparison test (**d, f, g, e**). Survival curves were analyzed for *P*-value using log-rank (Mantel-Cox) test; *P*-values < 0.0332 were considered significant.

NTZ's effect on the PRRSV lifecycle (Supplementary Fig. 9a). Interestingly, pre-treatment with NTZ showed a stronger antiviral effect, and adding NTZ even 16 h post viral inoculation still significantly inhibited PRRSV proliferation (Supplementary Fig. 9b). We then investigated NTZ's impact on key stages of PRRSV lifecycle: adsorption, entry, replication, and release, finding that NTZ primarily affects PRRSV replication (Supplementary Fig. 9c). Identifying the target protein of NTZ became imperative. Using the isothermal shift assay (iTSA), we discovered 26 potential NTZ targets (Fig. 5a). Further, Cellular Thermal Shift Assay-Western Blot (CETSA-WB) temperature range (TR) experiment indicated Glycine amidinotransferase mitochondrial (GATM),

NMRAL1, and PRRSV proteins NSP1α and NSP12 as potential NTZ targets (Fig. 5b, c). The compound concentration range (CCR) experiment also supported these four proteins as potential NTZ targets (Fig. 5d).

## NTZ targeting NMRAL1
In these proteins, we separately purified the proteins of NSP1α, NSP12, and NMRAL1. We found that the affinity of NMRAL1 with TIZ was higher than that of NSP1α and NSP12 (Supplementary Fig. 10). We speculate that NMRAL1 may be the primary target of NTZ action. Overexpression of NMRAL1 significantly increased both PRRSV titers and mRNA levels (Fig. 6a and b), while siRNA-mediated silencing of NMRAL1 reduced

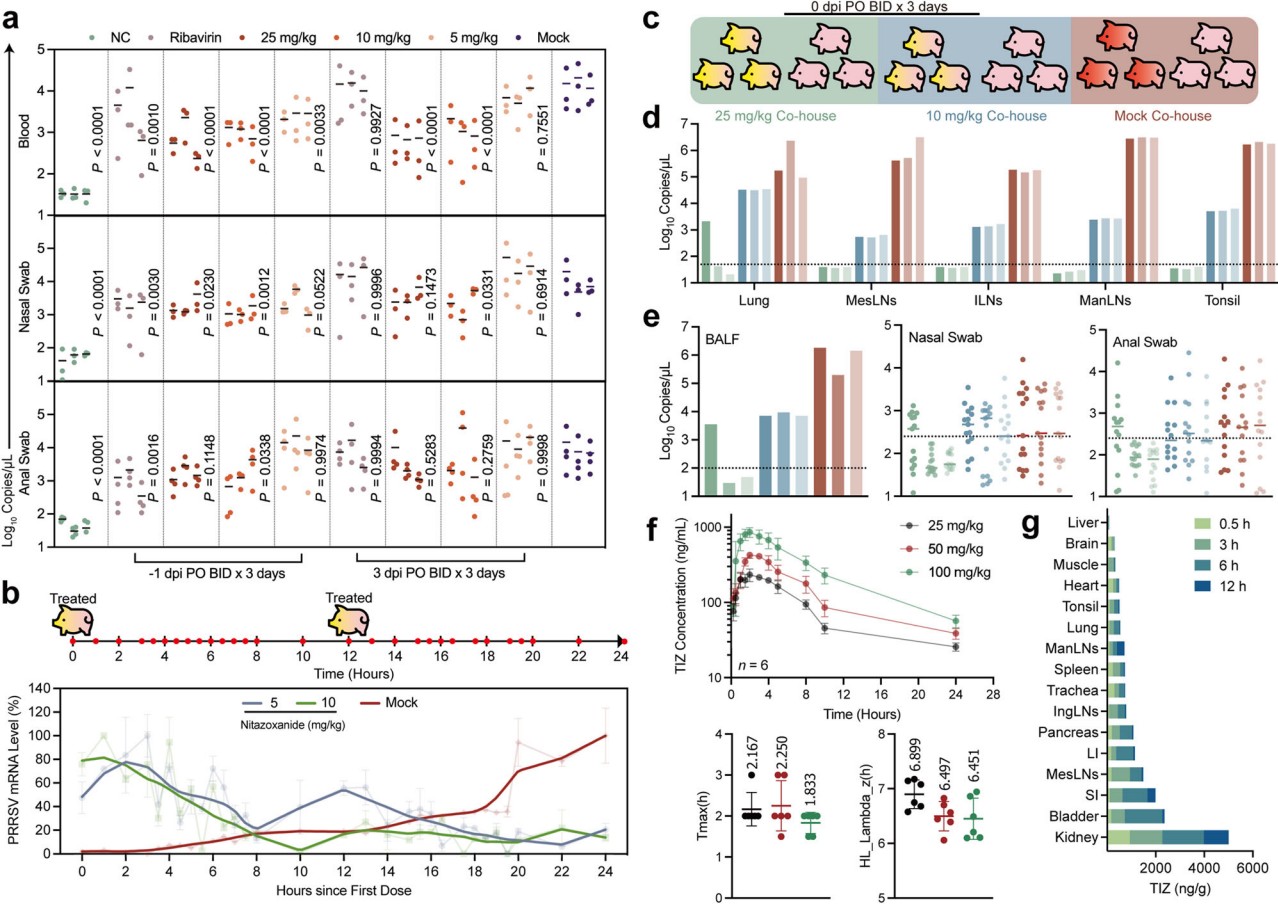

**Fig. 4 | NTZ inhibition of PRRSV shedding and transmission. a** Viral load in blood, and shedding in oronasal and anal swabs. Viral load or shedding in blood and oronasal swabs were quantified from 2 to 4 dpi, and in anal swabs from 2 to 5 dpi. Viral copy numbers for each pig during this period are presented. **b** NTZ suppression of PRRSV proliferation in blood. 48 h post infection (infected with $1 \times 10^{5.5}$ TCID$_{50}$ of GSWW-18 strain), pigs were orally administered NTZ at 10 mg/kg or 5 mg/kg at 0 h and 12 h, followed by blood collection via anterior vena cava or ear marginal vein. Data were normalized to highest and lowest values and fitted with LOWESS curves; semi-transparent lines represent pre-fitting data for each group. **c** Experimental design for NTZ reduction of PRRSV transmission. NTZ oral administration commenced simultaneously with infection ($1 \times 10^{5.5}$ TCID$_{50}$ of GSWW-18 strain) at 0 dpi, with daily collection of oronasal and anal swabs for 14 days, and all animals euthanized on day 14 for tissue and BALF collection. In the Mock co-house group, three animals survived until 14 dpi. **d** Viral loads in tissues of all experimental animals at 14 days. **e** Viral loads in BALF and shedding in oronasal and anal swabs over 14 days for PRRSV. **f** Pharmacokinetic curve and selected pharmacokinetic parameters. Parameters were calculated using a non-compartmental analysis (NCA). Eighteen pigs were evenly divided into three dosage groups ($n = 6$), each housed separately, fasted for 12 h before the experiment, and monitored by professional veterinarians for welfare and enclosure cleaning to prevent coprophagia. **g** Tissue distribution of NTZ's main metabolite TIZ. Tissues were collected as per Method Pharmacokinetics and Tissue Distribution of NTZ, quantifying TIZ content at various times ($n = 2$). Each bar represents an individual pig, with colors corresponding to the groups in Fig 4c and dashed lines represent the limit of detection (**d, e**). Symbols represent three or six independent biological replicates (**a, b, f**) Scatter plots (**a, f**) show mean ± SD values of samples, with $P$-values as indicated, calculated via two-way ANOVA with Dunnett's multiple-comparison test (**a**), $P$-values < 0.0332 were considered significant.

them (Fig. 6c, d). It was also observed that overexpression of NMRAL1 could not be inhibited by 10 µM NTZ, whereas siRNA transfection of NMRAL1 coupled with NTZ drastically reduced PRRSV-N protein levels (Fig. 6k). Previous studies have shown NMRAL1 inhibits retinoic acid-inducible gene I (RIG-I) mediated antiviral responses[24]. NTZ has been reported to activate the RIG-I pathway and induce an increase in IFN-β[25], which our findings corroborate (Fig. 6e–j). We discovered that NTZ addition increased NMRAL1 dimer formation (Figs. 5b, 6l), confirmed with purified NMRAL1 protein in vitro (Fig. 6m). Differential Scanning Fluorimetry (DSF) confirmed NTZ binding to NMRAL1 (Fig. 6n). Surface Plasmon Resonance (SPR) revealed the affinity of NTZ's primary metabolite TIZ for NMRAL1 to be -2.872 µM (Fig. 6o). Our molecular docking results showed TIZ forming hydrogen bonds with amino acid residues of both A and B chains of the NMRAL1 dimer, with a binding free energy of −5.82 kcal/mol (Supplementary Fig. 11, Supplementary Video 1). Molecular dynamics (MD) simulations were conducted for a more accurate understanding of TIZ's role within the NMRAL1 dimer.

Fluctuations or reductions in root-mean-square deviation (RMSD), Rg, and hydrogen bond count data after 80 ns indicated significant changes in the NMRAL1 system (Supplementary Fig. 12 a, b, d). The surface solvent accessibility (AREA) results suggested NMRAL1 area was on an increasing trend throughout the simulation, indicating dimer dissociation (Supplementary Fig. 12c). TIZ primarily interacted with A chain's LYS41, LYS42, LYS45, and B chain's LYS92, LYS89, LYS95, GLN93, LEU96, LYS118, and ARG103, with B chain's LYS92 and LYS89 and A chain's LYS41 being the hotspots with binding energies greater than −1 kcal/mol (Supplementary Fig. 13). This could explain the decrease in root-mean-square fluctuation (RMSF) values in the A chain's 30-60 region and the B chain's 85-105 region (Supplementary Fig. 12e). The addition of TIZ resulted in NMRAL1 having lower energy and a more compact conformation (Supplementary Fig. 14a, b). As simulation time progressed, match RMSD continuously increased, reaching its maximum at 80 ns, indicating ongoing dissociation of the A and B chains of the NMRAL1 dimer (Supplementary Fig. 14c). In

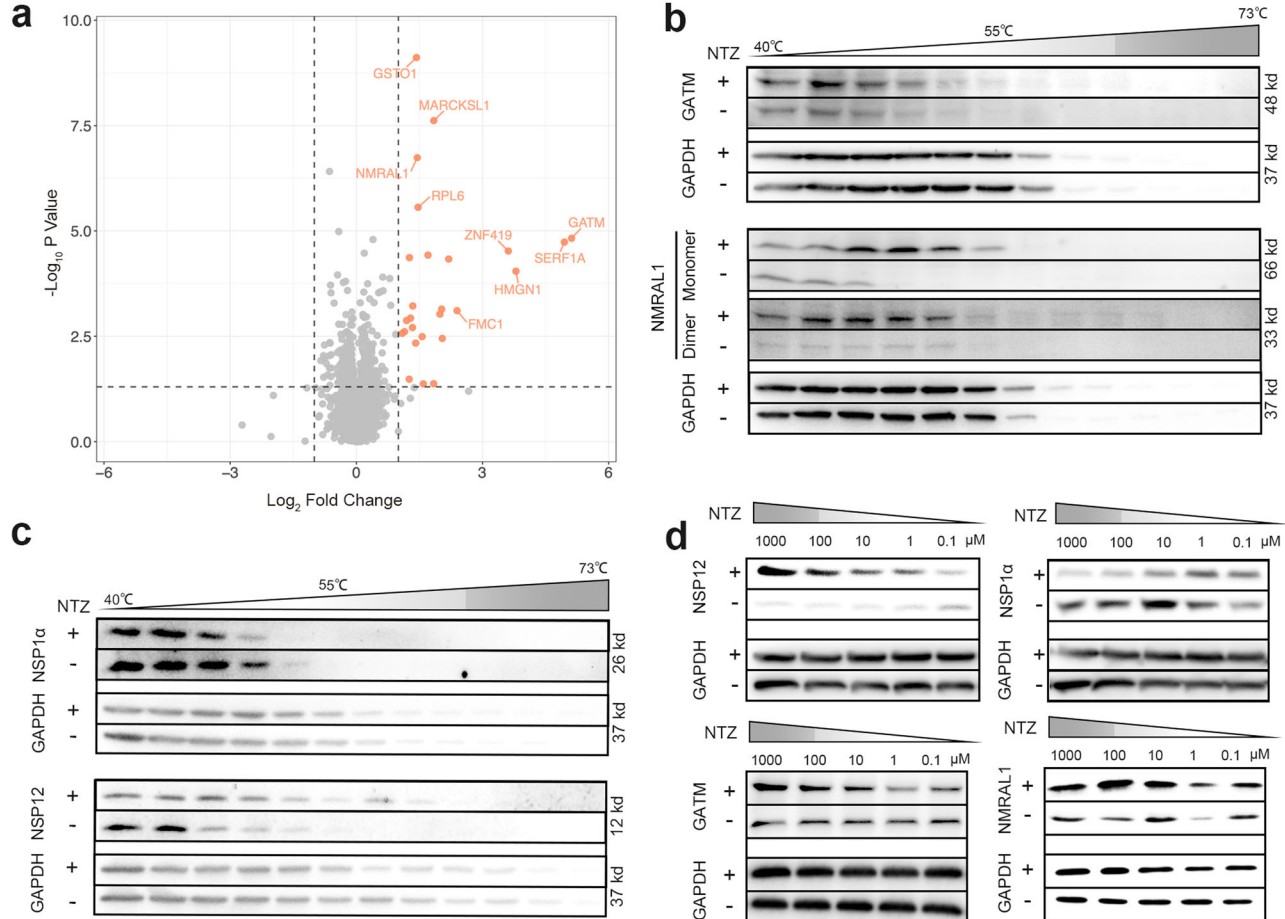

**Fig. 5 | NTZ targets multiple proteins. a** Potential NTZ targets identified by iTSA. Following Method Isothermal Shift Assay Proteome in Cell Lysates, iTSA was performed on cell lysates at 52 °C. With criteria set at Log$_2$ Fold Change > 1 and $P < 0.05$, a total of 26 potential targets were identified. The $p$-values were obtained from two-sample tests with empirical Bayes moderated $t$-statistics. **b** CETSA-WB validation of potential intracellular targets. Endogenous GATM and NMRAL1 in HEK-293T cells were verified as potential NTZ targets. GATM antibody (1:3000), NMRAL1 antibody (1:1500). **c** CETSA-WB validation of PRRSV non-structural protein targets. NSP1α and NSP12, plasmids of PRRSV non-structural proteins, transfected into Marc-145 cells, were identified as potential NTZ targets. HA antibody (1:5000). **d** Further verification of NTZ binding to aforementioned proteins using CCR method. All immunoblots had three biological repetitions, with similar results.

summary, MD data strongly suggests that TIZ can stabilize the conformation of the NMRAL1 dimer (Supplementary Videos 2 and 3).

## Discussion

The initial aim of our study was to find a strategy to control the PRRSV epidemic beyond vaccine immunization. While vaccine immunization undoubtedly remains a cornerstone in controlling viral outbreaks, its singular application has limitations in certain scenarios[26]. Thus, our focus pivoted towards the utilization of antiviral drugs, particularly in regions heavily afflicted by PRRSV. The aim was to aid in converting farms with unstable positive statuses into stable ones, thereby complementing the role of vaccines. To expedite the identification of effective antivirals against PRRSV, we devised a HTS methodology and analyzed 3274 compounds from the FDA-approved and pharmacopeial drug library. This process led to the identification of NTZ as a promising candidate. NTZ, an FDA-approved oral agent primarily used for protozoal diarrhea, exhibits minimal side effects in various species including mice, cats, and humans[21,27]. Our findings suggest its potential for rapid repositioning as a therapy for PRRS. Fifteen other identified compounds may serve as tool compounds or for combination therapy. These findings merit further exploration to ascertain their efficacy and applicability in PRRS management strategies.

As advocated by Garry P. Nolan, the maxim "Drug screening is best performed early using primary cells" holds particularly true in the

context of antiviral research[28]. Acknowledging this principle, and considering the potential limitations of our Marc-145-GFP cell line in fully mirroring the antiviral effects in primary cells, we employed unmodified Marc-145 cells and PAM cells sourced from BALF in the subsequent rounds of our screening process. The current absence of commercially available, specific therapeutic drugs and potent monoclonal antibodies against PRRSV, coupled with the limited neutralizing capacity of our laboratory-produced high-immune serum against PRRSV in vitro, necessitated the use of RBV as a positive control. This choice, despite RBV's recognized limitations in vivo (Figs. 3 and 4), underscores the critical need for identifying efficacious antiviral agents against PRRSV. Given the substantial global health risks posed by zoonotic viral transmissions from animals to humans, the urgency to discover and develop antiviral agents starting from animal pathogens is paramount. This approach is not only vital for controlling diseases within animal populations but also crucial in preventing potential spillovers to human populations[29].

In our study, NTZ demonstrated a notable capacity to alleviate severe pulmonary lesions induced by PRRSV, effectively reducing both viral shedding and the tissue viral load in PRRSV-infected animals. The oral administration of NTZ was observed to significantly reduce mortality rates among infected subjects. Importantly, NTZ impeded PRRSV-induced viremia, although its pulmonary distribution was somewhat limited, predominantly accumulating in lymph nodes and exhibiting a

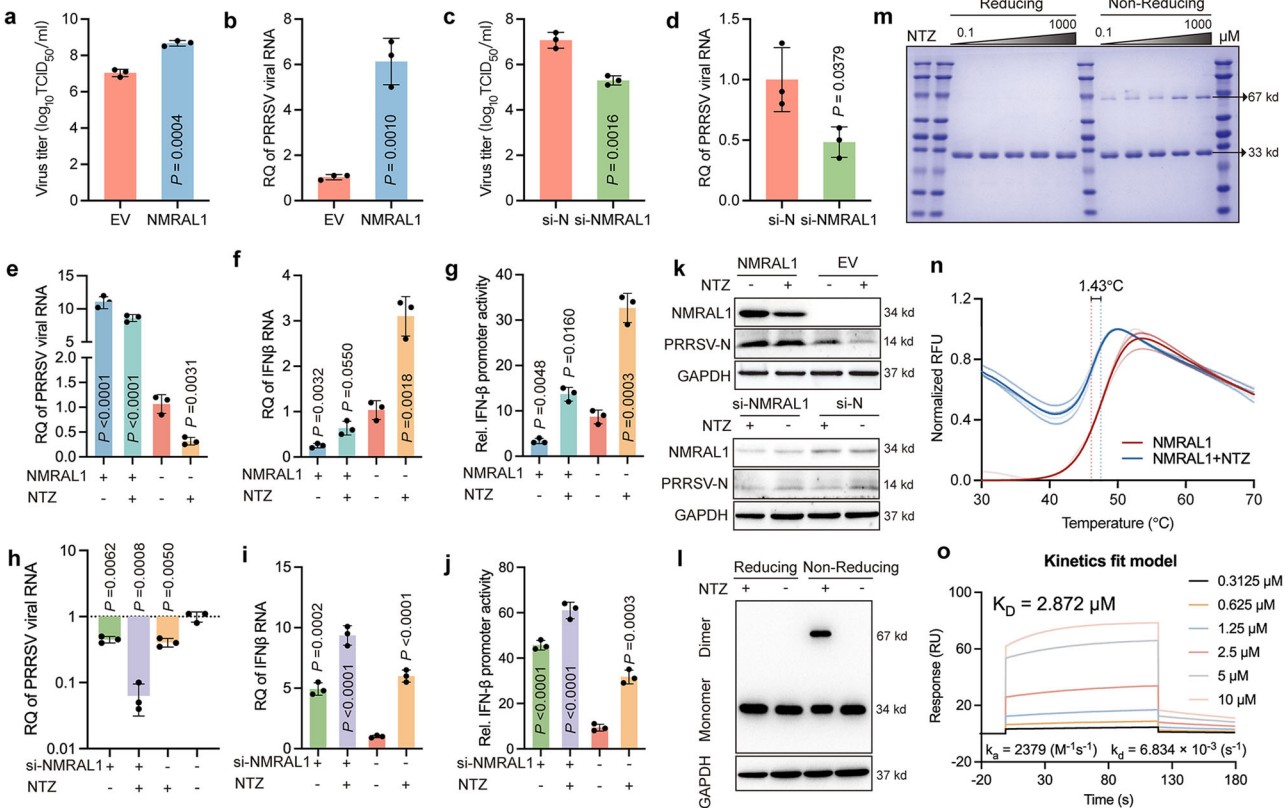

**Fig. 6 | TIZ enhances IFN-β pathway sensitivity by promoting NMRAL1 dimerization.** **a**, **b** NMRAL1 expression augmented VR-2332 viral titers and mRNA levels. **c**, **d** Silencing of NMRAL1 led to decreased VR-2332 viral titers and mRNA expression. **e** NMRAL1 attenuated the antiviral efficacy of NTZ. **f**, **g** NMRAL1 reduced the enhancement of IFN-β by NTZ. **h** NMRAL1 silencing intensified NTZ's viral inhibition. **i**, **j** NMRAL1 silencing further boosted NTZ's induction of IFN-β expression. **k** NMRAL1 expression enhanced PRRSV-N protein levels while diminishing NTZ's inhibitory effect on PRRSV-N. PRRSV-N antibody (1:1500), NMRAL1 antibody (1:1500). Silencing of NMRAL1 validated the same outcome. **l** Addition of NTZ increased intracellular NMRAL1 dimerization. HA antibody (1:5000). **m** In vitro purified NMRAL1 dimerization increased with higher NTZ concentrations. **n** DSF confirmed interaction between in vitro purified NMRAL1 and NTZ, increasing NMRAL1's Tm by 1.43 °C. **o** SPR revealed NMRAL1 binding to TIZ with an affinity $K_D$ of 4.505 µM (Supplementary Fig. 10a). NMRAL1 gene was synthesized and cloned into the PCDNA3.1 vector. Symbols represent independent biological replicates (**a**–**j**), showing mean values of samples, with P-values indicated, calculated via t-test (**a**–**d**) or two-way ANOVA with Dunnett's multiple-comparison test (**e**–**j**); P-values < 0.0332 were considered significant. All experiments were performed in triplicate and repeated at least three times. Data displayed are Mean ± SD from a representative experiment.

brief half-life. This distribution challenge could potentially be addressed by creating aerosolized particles of NTZ[30]. Prophylactic administration of NTZ was found to be more effective compared to its therapeutic use. Treatment was initiated on the third-day post-infection, coinciding with the onset of pronounced symptoms like elevated body temperature and viremia. Careful consideration is needed for using NTZ to control PRRSV spread, as the dosage may require further optimization, indicated by our finding that one out of three animals was infected in the 25 mg/kg NTZ group. Our previous incidental discovery of NTZ's inhibition of FCV proliferation in cats[21], and its reaffirmed efficacy against PRRSV in this study, has piqued our interest in its antiviral mechanism.

The multifaceted mechanism of action of NTZ, recognized as a broad-spectrum antiviral agent[31], presents a fascinating area of study. Utilizing iTSA, our research identified 26 potential target proteins for NTZ, inclusive of PRRSV non-structural proteins NSP1α and NSP12. This finding suggests that NTZ's antiviral impact may be both direct and indirect. Intriguingly, NMRAL1 (also known as HSCARG), an NADPH sensor responding to cellular oxidative stress and influencing pathways like RIG-I and NF-κB, has been identified as a key player in this context[32]. NMRAL1 monomers are known to inhibit TRAF3 ubiquitination, thereby attenuating the RIG-I pathway[24]. Our research confirmed that TIZ, the primary metabolite of NTZ, induces dimerization of NMRAL1, a phenomenon we observed both in cells and with purified proteins. Molecular docking and dynamics simulations further

confirmed the stabilizing influence of TIZ on the NMRAL1 dimer structure. Based on these findings, we hypothesize that NTZ's enhancement of the RIG-I pathway sensitivity is attributable to its promotion of NMRAL1 dimerization, which in turn reduces the prevalence of inhibitory monomers within the cytosol.

In conclusion, our research identified 16 potential compounds through HTS that inhibit PRRSV proliferation. We confirmed that high doses of NTZ correlate with antiviral activity against PRRSV in animal models. Animal models serve as gold standards for testing, and we observed better antiviral efficacy in prophylactic experiments, suggesting NTZ's suitability for preventing potential infections. Additionally, detailed studies on NTZ's resistance are warranted to prevent potential resistance. We identified 26 potential cellular target proteins for NTZ and two non-structural proteins within PRRSV as targets, with NMRAL1 being one of its targets. Ultimately, this work's workflow and assays may aid in developing and evaluating similar compounds, providing evidence for NTZ as an indirect antiviral agent. Proven in vivo efficacy makes NTZ a potential cornerstone in future defenses against PRRSV epidemics.

## Methods
### Cells and viruses
MRAC-145 or HEK-293T cells were cultured at 37 °C in a 5% $CO_2$ incubator and grown in Dulbecco's modified Eagle's medium (DMEM,

Gibco) supplemented with 10% fetal bovine serum (FBS, Gibco). The strains GSWW-15 (GenBank: KX767091.1) and GSWW-18 (GenBank: OP764591.1) were isolated and conserved in our laboratory[33]. The PRRSV VR-2332 strain (GenBank: EF536003.1) was purchased from Boehringer-Ingelheim. PAM were isolated and conserved as previously described[34]. Sequences of 3′UTR and 5′UTR derived from VR-2332 strain and GFP sequence from pcDNA3.1-EGFP were synthesized and inserted into the pLenti6/V5-DEST vector via homologous recombination. Post-generation of recombinant lentivirus, Marc-145 cells were infected to establish the Marc-145-GFP polyclonal cell line. A stable Marc-145-GFP monoclonal cell line was then obtained by limited dilution in medium containing 1 μg/ml Blasticidin S (B9300, Solarbio). All genes, primers, and siRNA were synthesized by Sangon Biotech (Shanghai) Co., Ltd.

## Antibodies and vectors

Vectors pLenti6/V5-DEST (VT1476), pSPAX2 (VT1444), pMD2.G (VT1443), PET28a, and pCMV-HA (VT1710) were acquired from Youbio. The Luc-IFN-β and pTK-RL were bought from Yeasen. Poly (I:C) was procured from MERCK (P1530). Genes encoding non-structural proteins were previously constructed and preserved by our laboratory.

The following antibodies were used for IFA and IHC: Mouse anti-SR30 (Rtilab, SR30-A), Goat Anti-Mouse HRP (Bioss, bs-0368G-HRP), Goat Anti-Mouse FITC (Bioss, bs-0368G-FITC).

For protein immunoblotting: Rabbit anti-GATM antibody (HPA026077, Sigma), Rabbit anti-NMRAL1 antibody (K004351P, Solarbio), Mouse anti-HA antibody (26D11, Abmart), Mouse anti-GAPDH (60004, Proteintech), Rabbit anti-PRRSV-N (bs-41387R, Bioss), Goat Anti-Mouse HRP Conjugate (1705047, Bio-rad), Goat anti-Rabbit HRP Conjugate (1705046, Bio-rad).

## Compounds, chemical library, and drug screening

Nitazoxanide (N159057), Tizoxanide (T343906) and Ribavirin (R101754) were procured from Aladdin. The FDA Approved & Pharmacopeial Drug Library (HY-L066) was obtained from MCE (MedChemExpress). Compounds were first diluted to 20 μM in medium. Then, 100 μL of this solution was mixed with 50 μL of cells (about 1000 cells) and 50 μL of PRRSV (MOI = 1), and added to a 96-well cell culture plate (269787, Thermo). Cultured at 37 °C in a 5% $CO_2$ incubator, cell numbers and fluorescence were assessed after 36 h using the Cytell™ Cell Imaging System (29-0567-49, GE). The Z' factor and S/N ratio were calculated following established methods[35].

## Indirect immunofluorescence and flow cytometry

Post-virus infection, cells underwent immunofluorescence assays under previously established conditions[36], detailed in the figure legends. A Leica DMI6000 microscope was utilized. GFP fluorescence in Marc-145 or Marc-145-GFP cells, post-PRRSV infection for 36 h and subsequent trypsinization (0.25%, 25200072, Gibco), was quantified using CytoFLEX LX (Beckman).

## IFN-β dual-luciferase reporter assay and RNA interference

As described previously, HEK-293T cells in 24-well plates were transfected with Luc-IFN-β, pTK-RL, Poly (I:C), and other plasmids[24]. After 24 h, cells were harvested for dual-luciferase reporter assay using the Glomax system (Promega). For RNAi and regular plasmid transfection, 2 μg of plasmid or 100 pM of NMRAL1 siRNA (si-NMRAL1) or NC (si-N) was transfected in 6-well plates as per the instructions of jetPRIME® transfection reagent (Polyplus). Sequences for si-NMRAL1: 5′-GAACAU UCAAGGUUCGAGUU-3′; si-N: 5′-UUCUCCGAACGUGUCACGUU-3′.

## Enrichment analysis

Targets, signaling pathways, and mechanisms of action from the FDA Approved & Pharmacopeial Drug Library were annotated for compounds. Enrichment analysis was conducted using GSEA software (v4.2.3) in prerank mode. Enrichment results with $P < 0.05$ and FDR $q < 0.25$ were considered significant. Additional enrichment analysis was performed using the free online tool Metascape. As previously mentioned, $P$-values were generated using a one-sided hypergeometric test[37]. Multiple hypothesis testing values were adjusted using the Benjamini–Hochberg method. Heatmap enrichment analysis was performed using the online, free SRplot tool for cluster analysis of compound targets[38].

## Viral titer and RNA expression level determination

Viral solutions were serially diluted by a factor of 10. Samples containing varying concentrations of virus (100 μL each) were combined with DMEM containing 2% FBS (100 μL) and added to wells; control wells were virus-free. Plates were incubated at 37 °C in a 5% $CO_2$ humidified environment for four days, after which viral $TCID_{50}$ values were calculated using the Reed and Muench formula. Relative quantification qRT-PCR was used to assess PRRSV gene expression, normalized using the $2^{-\Delta\Delta Ct}$ method in Design & Analysis Software (v2.6.0). Absolute quantification qRT-PCR was used for viral copy number detection, with RNA concentration normalization prior to analysis, as described in our previous publication[21]. Primers used included: PRRSV 5′-CTAAGAGAGGTGGCCTGTCG′ and 5′-GAGACTCGGCATACAGCACA-3′; Marc-145-GAPDH 5′-CCTTCCGTGTCCCTACTGCCAAC-3′ and 5′-GACGCCTGCTTCACCACCTTCT-3′; PAM-GAPDH 5′-TCTGGCAAAGTG-GACATT-3′ and 5′-GGTGGAATCATACTGGAACA-3′; IFNβ 5′-CACGA-CAGCTCTTTCCATGA-3′ and 5′-AGCCAGTGCTCGATGAATCT-3′; Human-GAPDH 5′-GTCGTCGACAACGGCTCCG-3′ and 5′-ATTGTA-GAAGGTGTGGTGC-3′.

## Cytotoxicity assay

Cells were seeded in 96-well plates and grown in DMEM supplemented with 10% FBS. Upon reaching confluence, compounds were diluted to varying concentrations in 2% FBS-containing medium. Control wells contained DMEM with 0.5% DMSO. Treated cells were incubated at 37 °C in a 5% $CO_2$ humidified environment for 48 or 72 h, followed by cytotoxicity assessment as outlined in our previous work[21].

## In vivo antiviral effect of NTZ

All animal experiments were conducted according to protocols approved by the Lanzhou Veterinary Research Institute's Animal Ethics Committee. Pigs were housed in the institute's specific pathogen-free facility and cared for by professional veterinarians. To minimize animal use, we followed Jaykaran Charan's guidelines and used GPower to determine sample sizes[39], dividing animals into ten groups ($n = 3$). Forty-day-old Landrace pigs (11–13 kg) were obtained from a farm in Gansu Province. They tested negative for ASFV, CSFV, PCV, and PRRSV via specific PCR/RT-PCR and commercial ELISA kits. All procedures conformed to China's general requirements for animal experiments (GB/T 35823-2018), and every effort was made to minimize animal discomfort. Euthanasia was administered to alleviate suffering in cases where animals couldn't feed or stand independently and showed diminished reactions to their environment.

For assessing the in vivo antiviral efficacy of NTZ, animals were divided into prophylactic (−1 dpi PO BID for 3 days) and therapeutic groups (3 dpi PO BID for 3 days), alongside a NC and PBS mock-treated group (Mock). Dosing gradients (25 mg/kg, 10 mg/kg, 5 mg/kg) were based on previous experiments. Daily rectal temperatures and body weights were monitored, and clinical scores were recorded on days 0 and 7 (Supplementary Table 1). Post-euthanasia, various tissues, and BALF were collected under veterinary guidance.

To verify NTZ's effect on PRRSV proliferation in blood, pigs were infected with $1 \times 10^{5.5}$ $TCID_{50}$ of the GSWW-18 strain and orally administered NTZ (10 or 5 mg/kg) 48 h post infection. Blood samples were taken from the ear margin or anterior vena cava at different time points, and virus copy numbers were analyzed as described above.

## Complete blood count, biochemical analysis, and immunohistochemistry

Blood samples were collected from the anterior vena cava for complete blood count using a PE-6800VET analyzer (Prokan) and biochemical analysis using a VetTest Plasma Chemistry Analyzer (IDEXX). Reference ranges were obtained from Iowa State University College of Veterinary Medicine (https://vetmed.iastate.edu/vpath/services/diagnostic-services/clinical-pathology/testing-and-fees/reference-intervals). IHC were performed as previously described[36], involving 24-h incubation with SR30 antibody (1:400) and subsequent treatment with HRP-labeled secondary antibody.

## Pharmacokinetics and tissue distribution of NTZ

Adhering to guidelines from the National Medical Products Administration for pharmacological safety studies, we determined the sample sizes. Pigs in different dosage groups were housed separately. For tissue collection, tissues were cleansed with PBS, blotted dry with filter paper, weighed, and then 1 g was placed in a 2 mL tube and stored on dry ice. Blood samples (1–2 mL) were collected from the anterior vena cava within 2 min at the designated times, centrifuged immediately at $14,000 \times g$ for 1 min, and the plasma was quickly preserved on dry ice. Subsequently, all samples were stored at −80 °C.

For tissue analysis, 1 mL of pure methanol was added to the samples along with zirconium oxide grinding beads and homogenized for 5 min. After centrifugation at 4 °C at $14,000 \times g$ for 10 min, the supernatant was filtered through a 0.22 μm filter for analysis. For plasma samples, after protein precipitation with methanol and centrifugation at $10,000 \times g$ for 10 min, the supernatant was subjected to LC–MS/MS analysis using a TSQ Quantum triple quadrupole mass spectrometer and an UltiMate 3000 RS chromatograph (Thermo). The standard curve was prepared with TIZ working solution in methanol across four orders of magnitude, mixed with blank pig plasma from pigs never exposed to NTZ, and diluted tenfold. Quality control samples prepared from blank plasma ensured accuracy and precision. Chromatographic conditions utilized a $2.1 \times 150$ mm, 5 μm, ZORBAX StableBond C18 column (Agilent). The flow rate was 0.5 ml/min, with mobile phase A comprising 0.1% formic acid in water and mobile phase B being 0.1% formic acid in acetonitrile; the needle wash was methanol, and the column temperature was 40 °C. Nitrogen was used as the sheath, auxiliary, and collision gas; the capillary temperature was set at 350 °C. The spray voltage for ESI was (+/-) 4000 V, with detection by selected reaction monitoring in negative ion mode. Data acquisition time was 5 min.

Data were initially collected and processed using Xcalibur (Thermo Fisher). Pharmacokinetic parameters were calculated using Phoenix WinNonlin (version 8.2, Certara) and graphed using GraphPad Prism. Data on the distribution of NTZ in tissues were collected and analyzed using Microsoft Excel (version 16.7) and visualized using GraphPad Prism.

## Cellular thermal shift assay

Cell lysates were prepared as previously described[40]. Marc-145 cells or HEK-293T cells transfected with plasmids, at a density of $5 \times 10^7$, were digested with trypsin, washed twice with PBS, and then resuspended in 1.3 mL of lysis buffer. For the TR experiment, lysates were subjected to three freeze-thaw cycles in liquid nitrogen, each time resuspended by pipetting with a 26-gauge needle, followed by centrifugation at 4 °C at $14,000 \times g$ for 30 min. Supernatants were divided equally into two 2 mL tubes: one with a final concentration of 40 μM NTZ and the other with an equivalent volume of DMSO. Samples were mixed, incubated at room temperature for 10 min, and then aliquoted into a 96-well PCR plate (50 μL/well). Two gradient programs were set on a PCR thermocycler (96-Well Thermal Cycler, ABI)−Program 1 with temperatures of 42, 45, 48, 51, 54, and 57 °C, and Program 2 with 60, 63, 66, 69, and 72 °C. Lids were heated to 90 °C to prevent evaporation. Each program

ran for 3 min. The contents of the PCR plate were then transferred to 1.5 mL tubes and centrifuged at 4 °C at $14,000 \times g$ for 30 min. Supernatants were either reserved as is or mixed with 4× Loading buffer containing 0.05% β-mercaptoethanol (M6250, Merck) and heated. For the CCR experiment, a 50 mM stock solution of NZT in 1% DMSO lysis buffer was diluted to concentrations of 2000, 200, 20, 2, and 0.2 μM. These were mixed with protein samples in a 1:1 ratio (final volume 50 μL) and processed as above after heating.

## Isothermal shift assay proteome in cell lysates

For the iTSA, each sample included five technical replicates. Cell lysates were prepared and treated with NTZ. The lysates underwent enzymatic digestion and Tandem Mass Tag labeling as described earlier[41]. Peptides were separated using a NanoElute ultra-high performance liquid chromatography system. Mobile phase A consisted of a water solution with 0.1% formic acid and 2% acetonitrile, while mobile phase B was a mixture of water and acetonitrile with 0.1% formic acid. The flow rate was maintained at 450 nL/min. Post-separation, peptides were ionized in a capillary ion source and analyzed using the timsTOF Pro 2 (Bruker) mass spectrometer. The ion source voltage was set to 2.0 kV. Both parent ions and secondary fragments of peptides were detected and analyzed using Time-of-Flight (TOF). The data acquisition mode was set to data-independent parallel accumulation-serial fragmentation (dia-PASEF) with a primary mass spectrometry scan range of 100–1700 m/z. After each primary MS scan, ten PASEF scans were performed within the same m/z range. Post-run data analysis was performed using the Spectronaut (version 16.0) software, employing default parameters for database search (Chlorocebus sabaeus[https://www.ncbi.nlm.nih.gov/datasets/genome/GCF_000409795.2] and NC_001961[https://www.ncbi.nlm.nih.gov/nuccore/NC_001961.1] database). Precursor and protein FDR were set at 1%. Data were further analyzed using the SRplot online tool[38].

## SDS-PAGE and western blot analysis

For Western blot analysis, protein samples were heated at 95 °C for 8 min and then cooled to room temperature. Samples were then loaded onto a 12% polyacrylamide gel for Sodium Dodecyl Sulfate–Polyacrylamide Gel Electrophoresis (SDS-PAGE). Following electrophoresis, proteins were transferred to a Polyvinylidene fluoride (PVDF) membrane (Merck) and blocked with 5% non-fat milk at room temperature for an hour. The membrane was subsequently incubated with specific primary antibodies and HRP-conjugated secondary antibodies. The protein signals were detected using a chemiluminescent substrate (34577, Thermo) and visualized using the Gel Doc XR+ molecular imager (Bio-Rad).

## Differential scanning fluorimetry and surface plasmon resonance

For DSF, 20 μM purified NSP1α and NSP12 protein from the GSWW-18 strain, as well as NMRAL1 protein (Accession: A0A0D9R9W0) was thoroughly mixed with 20 μM SYPRO Orange (S6650, Thermo) in a buffer containing 10 mM HEPES and 200 mM NaCl at pH 7.20. An equal volume of 100 μM NTZ, dissolved in the same buffer, was added and the mixture was filtered through a 0.22 μm filter. For controls, an equivalent volume of DMSO was used. Each reaction had a final volume of 25 μL per well. DSF experiments were monitored using the QuantStudio™ 5 Real-Time PCR System (Thermo). Data were analyzed using DSFworld[42].

For SPR, the Biacore T200 system (Cytiva) was used to assess binding affinity and kinetics. The Series S Sensor Chip CM5 (BR100530, Cytiva) immobilized 13,000 resonance units (RU) of NMRAL1 (50 μg/ml, pH 4.0, 10 μl/min, 25 °C). TIZ (10 to 0.3125 μM) dissolved in a 5% DMSO running buffer (1.05 × HBS-EP+, Cytiva) was applied, with a solvent correction fluid gradient of 4.5–5.8% DMSO. Contact time was set to 120 s for binding and 300 s for dissociation. Post-experiment, data were

analyzed using Biacore T200 Evaluation Software 3.0 with Affinity Fit 1:1 Binding model.

## Molecular docking and dynamics simulation

For molecular docking, NMRAL1 (Accession: A0A0D9R9W0) structures were generated using Swissmodel. TIZ (CID: 394397). The structures for docking prepared using Autodock Tools-1.5.7, centered at coordinates (center $x = 70.461$, center $y = 33.808$, center $z = 108.314$). Docking was performed using Watvina [https://github.com/biocheming/watvina], setting the box size to a 50 Å cube with a spacing step of 0.375 and a maximum limit of 10,000 models. The genetic algorithm was employed for conformation sampling and scoring, selecting the best conformations based on docking scores and structural rationale.

For MD simulations, Gromacs-2023.2 was used to simulate conformational changes in NMRAL1 and NMRAL1 + TIZ. The Amber14SB and TIP3P models were employed as the force field and solvent model. The protein was enclosed in a cubic box with a 0.1 nm boundary, filled with water molecules and counter ions for neutralization. Energy minimization was conducted for 50,000 steps using the steep method with a force convergence criterion of 1000 kJ/mol/nm, followed by NVT and NPT equilibrations for 100 ps each at 310.15 K and 1 bar pressure. Subsequent MD simulations utilized V-rescale thermostats and Parrinello–Rahman barostats. Post-simulation, the trajectories were analyzed using Gromacs for secondary structure, RMSD, RMSF, radius of gyration, solvent-accessible surface area, and hydrogen bonding. Data were processed or visualized using UCSF Chimera (1.17.3), UCSF ChimeraX (1.5), and Graphpad Prism.

## Figures and statistical analysis

Graphical representations and statistical analyses were primarily conducted using Graphpad Prism (version 9.5.1), except where otherwise specified. For comparisons involving more than two groups or two independent variables, one-way or two-way analysis of variance (ANOVA) with Dunnett's or Tukey's multiple comparisons post-tests were employed to assess statistical significance. For comparisons between two variables, two-tailed unpaired $t$-tests were used to determine statistical significance. Detailed statistical information for each experiment is described in the figure legends and corresponding Supplementary data. Final figures were assembled using Adobe Illustrator (CS6 version).

## Reporting summary

Further information on research design is available in the Nature Portfolio Reporting Summary linked to this article.

# Data availability

All data generated in this study are provided in the Supplementary Information/Source Data file. The MS raw files and proteome sequences data used in this study have been deposited to in the Proteome Xchange Consortium under accession code PXD050369. Additionally, the molecular dynamics trajectories data have been made publicly accessible via the public repository ScienceDB and are available here: https://doi.org/10.57760/sciencedb.18476. Source data are provided with this paper.

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

## Acknowledgements

This research was financially supported by the Agricultural Science and Technology Innovation Program of CAAS (CAAS-ASTIP-2022-LVRI), the Project of National Center of Technology Innovation for Pigs (grant no. NCTIP-XD/C03), the Key Development and Research Foundation of Gansu Province (22ZD6NA001), the Youth projects of Natural Science foundation of Gansu province(20JR10RA022), the Research Program of Lanzhou City, Gansu Province of China (2023-1-44). C.X. is member of the International Max Planck Research School for Genome Science.

## Author contributions

Conceptualization, Z.D.C.; methodology, Z.D.C., J.L.L., C.X., T.W., P.S. and J.L.W. formal analysis, Z.D.C., J.Y.L., G.X.L., J.C.Q., Y.Z., D.L.L., Y.S. and J.B.Y.; investigation, Z.D.C., J.L.L., C.X., K.L., Z.X.Z., H.Y., X.W.B., X.Q.M., P.H.L., Y.F.F., H.F.B., D.L., Q.Z., Z.X.L. and Y.M.C.; resources, J.C.Q., K.L., Z.X.Z., H.Y., X.W.B., X.Q.M., P.H.L., Y.F.F., H.F.B., D.L., Q.Z., Z.X.L. and Y.M.C.; writing—original draft, Z.D.C., C.X; writing—review & editing, Z.D.C., C.X., K.L, Z.X.Z., H.Y., X.W.B., X.Q.M., P.H.L., Y.F.F., H.F.B., D.L., Q.Z., Z.X.L., Y.M.C., J.Z. and Z.J.L.; visualization, Z.D.C., J.L.L., C.X., T.W., J.C.Q. and Z.J.L.; funding acquisition, J.Z. and Z.J.L.

## Competing interests

The authors declare no competing interests.
