## [Peer Review File · Nature Communications]

REVIEWER COMMENTS

Reviewer #1 (Remarks to the Author):

NCOMMS-23-59334A-Z: High-throughput screening unveils nitazoxanide as a potent PRRSV inhibitor by targeting NMRAL1.

Author Cui et al., presents extensive in vitro and in vivo experimental data that demonstrates Nitazoxanide as an inhibitor of PRRSV replication, transmission, infection and lesions associated with disease. The author provides an exhaustive array of data that supports NTZ as an inhibitor of NMRAL1 through dimerization supported by the metabolite TIZ. The manuscript is well-written and results support the conclusions of the author. The following are minor concerns for the author to consider.

Results:

1. Page 4, line 112: Please clarify the acronyms NC and RBV groups.
2. Page 4, line 105: What is meant by the author “these cells were also effective against two other PRRSV strains. . . .”? What were they effective against? Perhaps the sentence needs re-wording to clarify.
3. Page 8, line 190: The author refers to lung positive granules with PRRSV IHC. What are the granules? This seems an odd descriptive term. The IHC positive would indicate infected cells in the opinion of the reviewer. Please clarify.
4. Page 8, line 191: The images in Fig. 3c do not demonstrate much for IHC signals, particularly in the Mock group. These are low magnification images. Perhaps higher magnification of sections could demonstrate the PRRSV infected cells. At this magnification, it is not very convincing.
5. Page 8, line 196: Please review extended data figure 4a, it is unclear what the “collected organizations” represent. Please clarify if this should represent “tissue collection”.
6. Page 8, line 199-201: The author states only the MesLN had significantly suppressed virus in the treatment group. Considering this is Fig. 3f, it appears virus was significantly suppressed in the lung, ILN, ManLN, and Tonsil at 25 mg/kg compared to mock pigs. Please clarify.
7. Page 8, line 201-202: Is the author referring to the treatment group, prophylactic group, or both with this statement? It appears there were some scores for clinical signs in the prophylactic group treated with 25 mg/kg in Fig. 4e, correct?

8. Page 9, Fig 4e: The -1 dpi PO BID x 3 days (prophylactic) fig 4e Score graph does not follow the days post infection of 0 and 7 per each group. There are extraneous bars for the first two groups? Note that the 3 dpi PO BID x 3 days group (treatment) looks consistent with the day 0 and 7 with the different groups. This is lacking in the prophylactic group. Please ensure that Fig 4e is correctly presented.

9. Page 8, line 202-203: Again, is the author referring only to the treatment group with this statement or both groups? Please note that Fig. 3f is referenced for this sentence, but if BALF is the sample type, that is represented in Fig. 3g. It appears in the graph that for the prophylactic group, 10 and 25 mg/kg levels were significant, but only the 25 mg/kg in the treatment group. Please clarify.

10. Page 10, lines 246-248: Looking at Fig. 4b, it would appear the rapid decrease in PRRSV mRNA copy numbers occurred in the first 8-10 hours, then increased before decreasing again up to 24 hours. Please clarify.

Discussion:

11. Page 16, line 409-410: This sentence appears to repeat the statement in the sentence immediately prior. Please clarify if redundant.

Reviewer #2 (Remarks to the Author):

IN the manuscript NCOMMS-23-59334A-Z entitled High-Throughput Screening Unveils Nitazoxanide as a Potent PRRSV Inhibitor by Targeting NMRAL1 Cui and collaborators performed screening of over 3,200 compounds for their antiviral effects against the arterivirus PRRSV which causes significant disease outbreaks in swine. The authors then went on and characterized NTZ for its in vitro and in vivo antiviral efficacy against PRRSV. They performed comprehensive studies that demonstrate the antiviral activity of NTZ against PRRSV. Mechanistically the authors demonstrated that the primary metabolite of NTZ, Tizonamide (TIZ), targets the host protein NMRAL1 inducing its dimerization, which presumably leads to enhanced IFN-beta responses and an indirect antiviral effect. Overall the study is well written and add an important tool to control PRRRSV infections in the swine population. I have one major point that I think the authors need to address before the manuscript can be published.

1. Based on the iTSA screening the authors identified 26 potential targets for NTZ, including viral proteins nsp1alpha and nsp12. However, no further follow up on these proteins is presented in the

paper. I think it would be important for the authors to follow up on these two viral targets and characterize the mechanism of action of NTZ on these proteins.

Point-to-Point Response

Reviewer #1 (Remarks to the Author):

Comment:

NCOMMS-23-59334A-Z: High-throughput screening unveils nitazoxanide as a potent PRRSV inhibitor by targeting NMRAL1.

Author Cui et al., presents extensive in vitro and in vivo experimental data that demonstrates Nitazoxanide as an inhibitor of PRRSV replication, transmission, infection and lesions associated with disease. The author provides an exhaustive array of data that supports NTZ as an inhibitor of NMRAL1 through dimerization supported by the metabolite TIZ. The manuscript is well-written and results support the conclusions of the author. The following are minor concerns for the author to consider.

Response: We express our gratitude to the reviewers for their recognition of our work. We also appreciate the insightful comments and suggestions offered by the reviewers to enhance the quality of our manuscript. In response to these comments, we have re-supplied the immunohistochemistry (IHC) images and revised the manuscript to clarify data analysis, and interpretation to improve rigor. The revised sentence is shown in **red** below.

Results:

1. Page 4, line 112: Please clarify the acronyms NC and RBV groups.

Response: Thank you for the suggestion. We have included the full English names for NC and RBV.

*"Next, we selected the highest percentage of fluorescent cells in NC (**Negative Control**) and RBV (**Ribavirin positive control**) groups as the upper limit for antiviral activity (**Extended Data Fig. 1d**) and the average cell count in the RBV group as the lower limit for drug toxicity (**Extended Data Fig. 1e**). "*

2. Page 4, line 105: What is meant by the author "these cells were also effective against two other PRRSV strains. . . "? What were they effective against? Perhaps the sentence needs re-wording to clarify.

Response: Thank you for the suggestion. We sincerely apologize for the oversight and

the unclear description provided. We have revised the wording to further clarify that the GSWW-15, GSWW-18, and VR-2332 strains can produce green fluorescence when infecting Marc-145-GFP cells. The revised sentence is shown in red below.

" Upon infection of Marc-145-GFP cells with either the GSWW-15 or VR-2332 strains, green fluorescence emerged at the cytopathic effect (CPE) sites (Extended Data Fig. 1a)."

3. Page 8, line 190: The author refers to lung positive granules with PRRSV IHC. What are the granules? This seems an odd descriptive term. The IHC positive would indicate infected cells in the opinion of the reviewer. Please clarify.

Response: Thank you for the insightful suggestions. We sincerely apologize for the oversight and the unclear description provided. We have revised the phrasing of this sentence to clarify the IHC depiction.

"Immunohistochemistry (IHC) revealed that oral administration of 25mg/kg NTZ in both prophylactic and therapeutic groups resulted in fewer lung PRRSV-positive cells compared to MOCK-treated controls (Fig. 3c)"

4. Page 8, line 191: The images in Fig. 3c do not demonstrate much for IHC signals, particularly in the Mock group. These are low magnification images. Perhaps higher magnification of sections could demonstrate the PRRSV infected cells. At this magnification, it is not very convincing.

Response: Thank you for the insightful suggestions. We fully concur with the reviewer's suggestion. Consequently, we have replaced the images in Fig. 3c from the original 200x (Fig. R1a) with 400x magnification (Fig. R1b) to better demonstrate the IHC signals.

Fig. R1 IHC images at 200x **(a)** and subsequent re-captured IHC images at the same location under 400x **(b)**.

Fig. R1 b has been updated as **Fig. 3c** in the revised manuscript.

5. Page 8, line 196: Please review extended data figure 4a, it is unclear what the “collected organizations” represent. Please clarify if this should represent “tissue collection”.

Response: Thank you for the insightful suggestions. We deeply appreciate the reviewer's meticulous examination and sincerely apologize for the oversight. We have revised the wording in accordance with your recommendations.

Fig. R2 Animal experimental timeline. (a) Before revision: Extended Data Fig. 4 a. (b) After rephrasing: **Extended Data Fig. 4 a**.

Fig. R2 b has been updated as **Extended Data Fig. 4 a** in the revised manuscript.

6. Page 8, line 199-201: *The author states only the MesLN had significantly suppressed virus in the treatment group. Considering this is Fig. 3f, it appears virus was significantly suppressed in the lung, ILN, ManLN, and Tonsil at 25 mg/kg compared to mock pigs. Please clarify.*

Response: Thank you for the great suggestion. We deeply appreciate the reviewer's meticulous examination and sincerely apologize for the oversight. We mistakenly referred to ILNs as MesLNs. We extend our sincerest apologies for this error.

*"Notably, only the 25mg/kg dose significantly suppressed the virus in **ILNs** in the treatment group ($P < 0.0001$)."*

7. Page 8, line 201-202: *Is the author referring to the treatment group, prophylactic*

group, or both with this statement? It appears there were some scores for clinical signs in the prophylactic group treated with 25 mg/kg in Fig. 4e, correct?

Response: Thank you for the insightful suggestions. We deeply apologize for our mistake. Due to our oversight, the colors in the clinical score image for the prophylactic group were inverted relative to the actual dosages. We have revised the image (**Fig. R3 b**) and also updated the descriptions of clinical symptoms and mortality rates for both the treatment and prophylactic groups to clarify this issue.

"In both the treatment and prophylactic groups, the 25mg/kg group exhibited no apparent clinical symptoms by day 7 and no fatalities occurred within 14 days (Fig. 3e)."

Fig. R3 Clinical scores (left) and survival curves (right). **(a)** Before revision: **Fig. 3 e**. The section highlighted by the red box in the figure represents the portion that was previously described inaccurately. **(b)** After rephrasing: **Fig. 3 e**.

Fig. R3 b has been updated as **Fig. 3 e** in the revised manuscript.

8. Page 9, Fig 4e: The -1 dpi PO BID x 3 days (prophylactic) fig 4e Score graph does not follow the days post infection of 0 and 7 per each group. There are extraneous bars for the first two groups? Note that the 3 dpi PO BID x 3 days group (treatment) looks consistent with the day 0 and 7 with the different groups. This is lacking in the prophylactic group. Please ensure that Fig 4e is correctly presented.

Response: Thank you for the insightful suggestions. We extend our gratitude once again for your meticulous review. As indicated in **Fig. R3**, we have revised **Fig. 3e**. We sincerely apologize for any confusion caused by the discrepancies in **Fig. 3e**.

9. Page 8, line 202-203: Again, is the author referring only to the treatment group with this statement or both groups? Please note that Fig. 3f is referenced for this sentence, but if BALF is the sample type, that is represented in Fig. 3g. It appears in the graph that for the prophylactic group, 10 and 25 mg/kg levels were significant, but only the 25 mg/kg in the treatment group. Please clarify.

Response: Thank you for the great suggestion. We extend our gratitude once more for your detailed review. We sincerely apologize for any unclear descriptions previously provided. In the revised manuscript, we have rearticulated the depiction of Fig. 3f to enhance clarity.

*" In the bronchoalveolar lavage fluid (BALF), significant effects were observed in the prophylactic group at both the 10 mg/kg and 25 mg/kg dosages ($P < 0.0001$, **Fig. 3g**). However, in the treatment group, only the 25 mg/kg dosage achieved statistical significance ($P < 0.0001$, **Fig. 3g**)."*

10. Page 10, lines 246-248: Looking at Fig. 4b, it would appear the rapid decrease in PRRSV mRNA copy numbers occurred in the first 8-10 hours, then increased before decreasing again up to 24 hours. Please clarify.

Response: Thank you for the great suggestion. We have revised the description of Fig. 4b to further clarify the impact of NTZ on the proliferation of PRRSV in the blood.

" On the second day post-infection, we administered NTZ at dosages of 10 mg/kg or 5 mg/kg and monitored the dynamics of PRRSV in the blood within 24 hours (Fig. 4b). We observed that NTZ acted swiftly. With the 5mg/kg dosage, viral loads decreased from 2 hours to 8 hours post-administration, rebounded at 12 hours before the second dose, and were suppressed again after re-administration; the 10mg/kg dosage showed more pronounced inhibition (Fig. 4b), indicating direct suppression of PRRSV proliferation in blood by NTZ. "

Discussion:

11. Page 16, line 409-410: This sentence appears to repeat the statement in the sentence immediately prior. Please clarify if redundant.

Response: Thank you for the insightful suggestions. We are grateful once more for your detailed examination. Indeed, the sentence in question within our manuscript redundantly echoed the preceding statement. We have removed this superfluous sentence in the revised version of our manuscript.

" This choice, despite RBV's recognized limitations in vivo (Figs. 3 and 4), underscores the critical need for identifying efficacious antiviral agents against PRRSV. ~~This underseores the importance of identifying antiviral agents against PRRSV.~~"

Reviewer #2 (Remarks to the Author):

In the manuscript NCOMMS-23-59334A-Z entitled High-Throughput Screening Unveils Nitazoxanide as a Potent PRRSV Inhibitor by Targeting NMRAL1 Cui and collaborators performed screening of over 3,200 compounds for their antiviral effects against the arterivirus PRRSV which causes significant disease outbreaks in swine. The

authors then went on and characterized NTZ for its in vitro and in vivo antiviral efficacy against PRRSV. They performed comprehensive studies that demonstrate the antiviral activity of NTZ against PRRSV. Mechanistically the authors demonstrated that the primary metabolite of NTZ, Tizonamide (TIZ), targets the host protein NMRAL1 inducing its dimerization, which presumably leads to enhanced IFN-beta responses and an indirect antiviral effect. Overall the study is well written and add an important tool to control PRRRSV infections in the swine population. I have one major point that I think the authors need to address before the manuscript can be published.

Response: We are thankful for the reviewers' acknowledgment of our work. We appreciate your insightful comments and suggestions. In light of these remarks, we have provided additional experimental results and extensively revised the manuscript to enhance its rigor.

1. Based on the iTSA screening the authors identified 26 potential targets for NTZ, including viral proteins nsp1alpha and nsp12. However, no further follow up on these proteins is presented in the paper. I think it would me important for the authors to follow up on these two viral targets and characterize the mechanism of action of NTZ on these proteins.

Response: Thank you for the insightful comments and great suggestions. Previous research has indicated that the NSP1 protein of PRRSV acts as an inhibitor of Type I interferon¹. NSP1 undergoes autocleavage into Nsp1 α and Nsp1 β through its own papain-like cysteine protease (PCP) domain². The N-terminal zinc finger motif (ZF1) of Nsp1 α is an essential element for its interferon inhibition capability². This may involve the nuclear shuttling of Nsp1 α facilitated by ZF1, as Nsp1 α cannot enter the nucleus when ZF1 is absent. NSP12 is a "mysterious" protein, whose functions have only recently begun to be unveiled³. NSP12 is likely associated with the synthesis of viral subgenomic mRNA, and PRRSV cannot replicate normally when NSP12 is disrupted or absent^{3,4}. However, to date, no crystal structure of NSP12 has been reported,

possibly due to its challenging expression in supernatants.

Indeed, our investigation into Nsp1 α and NSP12 as potential targets of NTZ commenced upon their discovery, but the lack of antibodies against Nsp1 α and NSP12, coupled with their scant biological research foundation, posed significant challenges. Our research into these proteins has been relentless, but due to unpredictable experimental timelines, their mechanisms were not fully elucidated in this paper. We apologize for any confusion this may have caused. Our intention is to further investigate Nsp1 α and NSP12 in future studies to thoroughly reveal their direct antiviral actions. To address the reviewers' concerns, we have endeavored to purify Nsp1 α (**Fig.R4 a**) and NSP12 proteins and conducted in vitro studies with TIZ, NTZ's primary metabolite (**Fig.R5 d**). Given that the T_m value of Nsp1 α protein, post-GST tag cleavage, differs from that in eukaryotic cells and NSP12 was refolded from inclusion bodies (IBs), the purified Nsp1 α and NSP12 might not reflect their true conformations.

We employed multiple strategies to express and purify NSP12, but the outcomes were consistently suboptimal. To date, we have been unable to obtain soluble NSP12 protein directly. We attempted using GST tags (**Fig. R4 b**), SUMO tags (data not shown), and HIS tags (**Fig. R4 c**), but NSP12 was invariably expressed in IBs. In the baculovirus expression system, the yield of HIS-tagged NSP12 was so low that it precluded purification (**Fig. R4 d**). Consequently, we were compelled to use NSP12 protein with a His tag, refolded from IBs, for subsequent experiments.

Fig.R4 Expression and Purification of Nsp1 α and NSP12 Proteins

(a) Purification of Nsp1 α protein with a GST tag. 'R' denotes the purified GST-Nsp1 α protein.

(b) Expression and identification of NSP12 protein with a GST tag. 'NC' represents the total bacterial lysate before induction. '1' and '2' are the total bacterial lysates after induction. 'NC1' is the supernatant without induction. '3' and '4' are the supernatants after induction. 'NC2' is the precipitate after induction. '5' and '6' are the precipitates after induction. The right image shows the results of the Western Blot (WB) analysis.

(c) Expression and identification of NSP12 protein with a His tag. '0' is the total bacterial lysate before induction. '1' is the total bacterial lysate after induction at 15 $^{\circ}$ C for 16 hours. '2' is the total bacterial lysate after induction at 37 $^{\circ}$ C for 16 hours. '3' is the supernatant after induction at 15 $^{\circ}$ C. '4' is the precipitate after induction at 15 $^{\circ}$ C. The WB analysis confirmed the presence of only trace amounts of NSP12 protein in the supernatant (middle image). No NSP12 protein was obtained from the supernatant using a Ni-IDA affinity chromatography column (right image).

(d) In the baculovirus expression system, expression and identification of NSP12 protein with a His tag. 'L1' is the total cell lysate. 'S1' is the supernatant after cell lysis.

'FT' is the flow-through after passing through the Ni column. 'Ni-200' and 'Ni-500' represent different purification conditions with varying concentrations. 'L2' is the total cell lysate after changing the expression conditions. 'S2' is the supernatant after cell lysis with changed expression conditions. The portions highlighted in red boxes indicate the target protein.

Additionally, our supplementary data (**Fig. R5**) revealed that the dissociation constant (K_D) between Nsp1 α and TIZ is lower than that of NMRAL1. We observe that NSP12 purified by IBs did not exhibit binding activity with NTZ in vitro, even when attempting different systems (**Fig. R5 e-h**). Although SPR confirms the interaction between NSP12 and TIZ, this interaction is notably weak (**Fig. R5 i**). This could possibly be attributed to the abnormal protein conformation of NSP12 purified by IBs. The existing data also indicate that the early addition of NTZ produces a more pronounced antiviral effect (**Extended Data Fig. 9**), and the overexpression of NMRAL1 can almost negate the antiviral action of NTZ. This leads us to speculate that NMRAL1 may be one of the primary targets of NTZ's mechanism of action, while Nsp1 α and NSP12 may serve as secondary targets.

Fig.R5 Validation of the binding between Nsp1 α and Nsp12 with TIZ through SPR, DSF or CETSA.

- (a) The K_D value of NMRAL1 obtained through the Affinity fit model.
- (b) The K_D value of Nsp1 α determined using the Kinetics fit model.
- (c) The K_D value of Nsp1 α obtained through the Affinity fit model.

- (d) DSF confirmed the interaction between in vitro purified Nsp1 α and NTZ, resulting in a decrease of 3.75°C in the T_m value of Nsp1 α .
- (e) In 10 mM HEPS and 200 mM NaCl (pH = 7.2), NSP12 did not exhibit the T_m value.
- (f) In 10 mM HEPS and 200 mM NaCl (pH = 7.2), NSP12 did not exhibit interaction with NTZ.
- (g) In PBS (pH = 7.4), NSP12 did not exhibit interaction with NTZ.
- (h) In PBS and 500 mM NaCl (pH = 7.4), NSP12 did not exhibit interaction with NTZ.
- (i) The K_D value of Nsp12 determined using the Kinetics fit model.

Fig. R5 a-e and i have been updated as **Extended Data Fig. 10** in the revised manuscript. Additionally, we simultaneously modified the **Results** section of the manuscript to further clarify that NMRAL1 may be the primary target of NTZ.

"In these proteins, we separately purified the proteins of NSP1 α , NSP12, and NMRAL1. We found that the affinity of NMRAL1 with TIZ was higher than that of NSP1 α and NSP12 (Extended Data Fig. 10). We speculate that NMRAL1 may be the primary target of NTZ action. Overexpression of NMRAL1 significantly increased both PRRSV titers and mRNA levels (Figs. 6a and b), while siRNA-mediated silencing of NMRAL1 reduced them (Figs. 6c and d)."

Reference

1. Kim, O., Sun, Y., Lai, F. W., Song, C. & Yoo, D. Modulation of type I interferon induction by porcine reproductive and respiratory syndrome virus and degradation of CREB-binding protein by non-structural protein 1 in MARC-145 and HeLa cells. *Virology* **402**, 315–326 (2010).
2. Han, M., Du, Y., Song, C. & Yoo, D. Degradation of CREB-binding protein and modulation of type I interferon induction by the zinc finger motif of the porcine reproductive and respiratory syndrome virus nsp1 α subunit. *Virus Research* **172**, 54–65 (2013).
3. Wang, T.-Y. *et al.* The Nsp12-coding region of type 2 PRRSV is required for viral subgenomic mRNA synthesis. *Emerging Microbes & Infections* **8**, 1501–1510 (2019).
4. Li, L. *et al.* PSMB1 Inhibits the Replication of Porcine Reproductive and Respiratory Syndrome Virus by Recruiting NBR1 To Degrade Nonstructural Protein 12 by Autophagy. *Journal of Virology* **97**, e01660-22 (2023).

REVIEWERS' COMMENTS

Reviewer #1 (Remarks to the Author):

Author Cui et al., provides an updated manuscript including edits requested by the reviewer. The manuscript is improved based on changes presented by the author as well as improvements to figures and tables. The manuscript is well-written and results support the conclusions of the author. The reviewer has no additional questions or concerns.

Reviewer #2 (Remarks to the Author):

The authors successfully addressed the review comments and the manuscript can be accepted.

Point-to-Point Response

Reviewer #1 (Remarks to the Author):

Comment:

Author Cui et al., provides an updated manuscript including edits requested by the reviewer. The manuscript is improved based on changes presented by the author as well as improvements to figures and tables. The manuscript is well-written and results support the conclusions of the author. The reviewer has no additional questions or concerns.

Response: We express our gratitude to the reviewers for their recognition of our work. We also appreciate the insightful comments and suggestions offered by the reviewers to enhance the quality of our manuscript.

Reviewer #2:

The authors successfully addressed the review comments and the manuscript can be accepted.

Response: We are thankful for the reviewers' acknowledgment of our work. We appreciate your insightful comments and suggestions.